# The Role of Macrophages in Liver Fibrosis: New Therapeutic Opportunities

**DOI:** 10.3390/ijms23126649

**Published:** 2022-06-14

**Authors:** Eleonora Binatti, Alessio Gerussi, Donatella Barisani, Pietro Invernizzi

**Affiliations:** 1Division of Gastroenterology, Center for Autoimmune Liver Diseases, Department of Medicine and Surgery, Università degli Studi di Milano Bicocca, 20900 Monza, Italy; alessio.gerussi@unimib.it (A.G.); pietro.invernizzi@unimib.it (P.I.); 2European Reference Network on Hepatological Diseases (ERN RARE-LIVER), San Gerardo Hospital, 20900 Monza, Italy; 3School of Medicine and Surgery, University of Milano-Bicocca, 20900 Monza, Italy; donatella.barisani@unimib.it

**Keywords:** fibrosis, liver fibrosis, inflammation, macrophages, Kupffer cells, drug delivery, targeted therapies

## Abstract

Chronic inflammation is the hallmark of fibrotic disorders and is characterized by the activation of immune cells in the damaged tissues. Macrophages have emerged as central players in the fibrotic process since they initiate, sustain and amplify the inflammatory reaction. As regards the liver, distinct populations of phagocytic cells, like Kupffer cells and monocyte-derived macrophages, are indisputably key cells implicated in the pathogenesis of several chronic liver diseases. In this review, we summarize the current knowledge on the origin, role and functions of macrophages in fibrotic conditions, with a specific focus on liver fibrosis; then, we discuss some innovative therapeutic strategies targeting macrophages in fibrotic liver diseases.

## 1. Introduction

Fibrotic disorders affect nearly all tissues and organ systems, especially liver, lungs, skin, bowel, kidneys, heart, and eyes. Fibrotic tissue responses are considered a leading cause of morbidity and mortality, and impose a major socio-economic burden on modern societies. In the United States, 45% of death can be attributed to fibrosis [1]. Its worldwide incidence and the associated health-care burden are spreading and, therefore, fibrosis is increasingly recognized as one of today’s major healthcare challenges [1].

Tissue fibrosis is defined as a wound healing response that has gone out of the control as a consequence of persistent inflammatory injuries to epithelial and endothelial tissues induced by different stimuli, e.g., infections, autoimmune reactions, allergic responses, exposure to radiations, toxins, chemical and mechanical insults. Recurrent or persistent epithelial and/or endothelial cell damage is a core element that initiates and sustains the progression of fibrosis. At least five responses to injury-induced functional or physical disruption of epithelial cells can provoke tissue fibrosis [2]: cell death, dysregulation of metabolic pathways that cause cell stress and activation, epithelial-to-mesenchymal transition (EMT), the interaction between integrins and the transforming growth factor-beta (TGF-β), and the involvement of innate and adaptive immune responses. These responses involve a complex multistage inflammatory process [2] orchestrated by a network of cytokines, chemokines, growth factors, adhesion molecules, and signaling processes. All these events end in an excessive accumulation and deposition of extracellular matrix (ECM) components, including mixture of proteins (collagens and elastin), glycoproteins and proteoglycans (fibronectin, laminin, and tenascin), glycosaminoglycans (heparin and chondroitin sulphates), that progressively remodel and destroy the normal tissue architecture, ultimately leading to its dysfunction and to an eventual organ failure.

One of the main actors involved in the development of this condition is represented by immune cells, in particular by macrophages. Macrophages are critical regulators of the whole fibrotic process, as they are responsible to initiate, sustain and amplify the detrimental inflammatory cascade, leading to fibrosis. Remarkable progresses have been achieved to elucidate the pathogenesis of fibrotic disorders, and to identify molecular pathways involved in these conditions. However, despite all these considerations, up to now there is no approved efficient treatments.

Here, we summarize the current knowledge of macrophages, their origin, their heterogeneity and their roles in inflammation and fibrosis. A better understanding of their functions in these conditions, and in details in liver fibrosis, would be of great potential therapeutic interest to prevent/reduce the progression of fibrosis. Then, we provide an overview of the most recent potential therapeutic strategies to target hepatic macrophages for the treatment of liver fibrosis.

## 2. Role of Macrophages in Fibrotic Disorders

It has been estimated that around 0.2 trillion of macrophages are present in almost every tissues and body compartments. Macrophages are widely distributed, either as resident cells or monocyte-derived cells that infiltrate into tissues of adult mammals [3], where they can represent up to 10–15% of the total cell number in quiescent conditions [4], and perform specialized functions. Macrophages are central players of the immune response and their canonical role is to continuously scan the tissue in which they reside, and actively participate in maintaining homeostasis and integrity [5]. It is not surprising that abnormal macrophage behavior has been implicated in the pathophysiology of several human disease conditions, including inflammatory and fibrotic conditions [5].

Following tissue injury, macrophages orchestrate the entire wound healing process by promoting inflammation and resolving it, by stimulating cell proliferation and tissue restoration. Uncontrolled process and disturbances in macrophages functions (e.g., an exacerbate inflammatory response) contribute to impairing healing and can lead to an aberrant repair and then to fibrosis. Therefore, it is fundamental to better understand the mechanisms that contribute to the development of fibrotic disease, consequently to macrophage response and activation, and to dissect macrophage functions in both physiological and pathological conditions.

### 2.1. Origin

Macrophage cells, first discovered in the late 19th century by the Nobel Prize laureate Ilya Metchnikoff, are white blood cells and their main functions are to phagocytize pathogens, cellular debris, death cells, tumor cells and to activate lymphocytes and other immune cells. Macrophages are evolutionary conserved phagocytes that evolved more than 500 million years ago. The prevailing view in the last 50 years, proposed by van Furth in the 1960s, was that macrophages mainly originated from circulating adult blood monocytes that differentiate from their precursor in the bone marrow [3]. However, current models have proposed that tissue resident macrophages have a different embryonic origin and persist into the adulthood as resident and self-maintaining populations. In the early stage of the embryonic process, macrophages are first detected in the extra-embryonic yolk sac (“primitive hematopoiesis”). During this stage, macrophages are the only produced white blood cells. Then, definitive hematopoietic stem cells (HSCs) spring from the aorta-gonad-mesonephros and originate all immune lineages. HSCs migrate to the fetal liver which serves as the major hematopoietic organ during the remaining embryonic development [3]. Yolk-sac-derived macrophages and definitive-HSC-derived macrophages differ in the transcription factor usage and in the expression of surface markers. Yolk-sac-derived progenitors are independent on the transcription factor MYB and express CXCR1^hi^, F4/80^hi^ CD11b^low^ [6], whereas definitive HSC-derived macrophages develop entirely dependent of MYB [7]. Unfortunately, many adult resident tissue macrophages that originate during embryonic development alter the expression of cell-surface markers as the animal matures, hindering the ability to precisely track macrophage populations [3]. After birth, macrophages persist into the adulthood as resident and self-maintaining populations and monocytes derived from bone marrow can replenish tissue resident macrophages following injuries, infections, and inflammations [8].

Respectively to the liver, Kupffer cells (KCs), the resident macrophages, originate in the yolk sac from colony-stimulating factor 1 receptor (CSF1R)^+^ erythromyeloid progenitors (EMPs) [9,10]. EMPs migrate to the nascent fetal liver around embryonic day (E) 10.5 in mice, and on E 10.5–12.5 EMPs mature into fetal liver monocytes which give rise to KCs. On the contrary, the circulating monocytes are the precursor of bone marrow monocyte-derived macrophages (MoMϕs, cells that rapidly accumulate and are activated following virtually any organ injury, to be distiguished from KCs, the resident population). Circulating monocytes are principally generated from a chemokine (C-X3-C motif) receptor 1 (CX3CR1)^+^ CD117^+^ (c-kit, the receptor for stem cell factor) lineage-negative (Lin^−^) bone marrow progenitor population [10]. Fogg D. K. et al., demonstrated that CX3CR1^+^ CD117^+^ Lin^−^ cells grown on S17 stroma possessed morphology typical of MΦs and dendritic cells (DCs) but not polymorphonuclear cells (PMN). In contrast, CX3CR1^−^ CD117^+^ Lin^−^ cells gave rise to colonies of PMN, MΦs, and DCs [11].

### 2.2. Different Populations of Macrophage

Macrophages take different names depending on their localization, e.g., alveolar macrophages in the lungs, KCs in the liver, microglia in the nervous system, Langerhans cells in the skin, and splenic red pulp macrophages in the spleen, as shown in Figure 1.

These populations of macrophages possess distinctive transcriptional profiles that allow them to be considered as many different and unique classes of macrophages. However, the functions of macrophages are basically the same in all tissues: they are pivotal players in tissue development and immune surveillance, they fight against and eliminate pathogens and other harmful organisms, and they contribute to the maintenance of tissue homeostasis.

Moreover, in the same tissue different subpopulation of macrophages can coexist, e.g., in the liver KCs and MoMϕs are distinct subsets of macrophages.

KCs represent the main hepatic macrophages during steady state involved in homeostasis. Metabolic or toxic damage results in massive infiltration of MoMϕs into the injured liver. The infiltrating macrophages MoMϕs are immunogenic in nature and receive signals from the local microenvironment that prompt their functional differentiation [12,13]. Morphologically, MoMϕs are relatively circular opposed to KCs that have long cytoplasmatic expansions allowing them to be sensing the microenvironments. KCs and MoMϕs can be further distinguished from each other based on their differential expression of cell surface markers [9]. Murine KCs are CD11b^low^, F4/80^high^, and Clec4F^+^; human KCs are CD68^+^, CD14^+^, TLR4^+^, and CXCR1^−^. While murine MoMϕs are CD11b^+^, F4/80^intermediate^, Ly6C^+^ and CSF1R^+^; human MoMϕs are CD14^+^, CCR2^+^, and CD16^+/−^ [9]. Alternatively KCs are distinguishable from MoMϕs based on their expression of the T cell immunoglobulin (Ig) and mucin domain containing 4 (timd4) and stabilin 2 (stab2) gene receptors [10]. Further, in mouse models of liver diseases, MoMϕs are divided into Ly6C^high^ and Ly6C^low^ MoMϕs, according to lymphocyte antigen 6 complex, locus C1 (Ly6C, previously termed Gr1) expression levels. In human liver, single-cell sequencing shed new light on different expression markers of intrahepatic macrophages that consist in CD68^+^MARCO^+^ KCs, CD68^+^MARCO^−^ macrophages, and CD14^+^ monocytes [9,10,14,15]. (See Appendix A for a general overview on macrophage markers and their functions).

### 2.3. Macrophages Functions, Heterogeneity, and Polarization

Different macrophages functionalities and activities, needed to keep homeostasis, reflect the differences in their origin, distribution, and changes in tissue micro-environments. The maintenance of homeostasis can be perturbed by chronic insults which cause an anomalous amplification of macrophages activities and consequently leads to a causal association between macrophages and diseases. Macrophages are equipped with a wide variety of receptors that recognize different stimuli and make them efficient to phagocytize and induce the production of inflammatory cytokines. Table 1 summarizes the functions of resident macrophages in some organs of human body in both physiological and pathological conditions [4,16,17,18]. It is not sufficient to classify macrophages in several populations, depending on their localization. To better describe the whole spectrum of these phagocytic cells, it is needed to take in consideration their different phenotypes in response to the different microenvironment conditions. Macrophages possess a considerable plasticity that allow them to efficiently respond to environmental signals and change their phenotype. Usually, early signals are typically generated by immune cells and can exert effects on the physiology of macrophages. Depending on their physiology consequently to the diverse activations, macrophages are categorized in phenotypically distinct subpopulations. Broadly speaking, a dichotomy has been proposed for macrophages activation: classic vs. alternative, also M1 and M2, respectively [19].

#### 2.3.1. Classically Activated Macrophages (M1)

In the 1960s, George B. Mackaness first introduced the term “macrophages classical activation” to report a microbicidal activity of macrophages toward the bacillus Calmette-Guerin (BCG) and Listeria upon secondary exposure to pathogen [19]. Mills and colleagues discovered that classically activated macrophages (M1) respond to interferon-gamma (IFNγ), lipopolysaccharide (LPS), granulocyte-macrophage colony stimulating factor (GM-CSF), and tumor necrosis factor (TNF) stimuli [19]. IFNγ is the main cytokine linked with M1 activation and an important Th1 response-mediator.

Natural killer (NK) cells are the principal source of IFNγ. Following an infection or stress, NK cells produce IFNγ that in turn primes macrophages to release pro-inflammatory cytokines (e.g., IL-1β, IL-12, IL-18, IL-23) and to produce high amount of reactive oxygen and nitrogen species (ROS and RNS, respectively), for example through the up-regulation of the NADPH oxidase and the inducible nitric oxide synthase (iNOS), respectively. This leads to drive antigen specific Th1 and Th17 cell inflammatory responses.

Classically activated macrophages phenotypically express high levels of major histocompatibility complex class II (MHC II), CD68, CD80 and CD86 (see Figure 2). They exhibit inflammatory functions and are implicated in initiating and sustaining inflammation in the context of the host defense, and if this process is not finely controlled, it is detrimental to the health of the host.

#### 2.3.2. Alternatively Activated Macrophages (M2)

In the 1990s it was discovered that interleukin-4 (IL-4) induced different effects on macrophage gene expression compared to that of IFNγ and LPS. In contrast to the classical activation of macrophages by IFNγ, this activation induced by IL-4 was described as “alternative activation” [20]. In particular, basophils and mast cells are important early producers of IL-4, but other granulocytes can also contribute. Fungal cells, immune complexes, helminth infections, apoptotic cells, IL-13, IL-10, TGF-β, and macrophage colony-stimulating factor (MCSF) can further induce alternative activation (see Figure 2). The early production of IL-4 converts resident macrophages into a population programmed to antagonize the inflammatory responses and markers, in particular IL-4 stimulates macrophages to switch on the arginase enzyme and to release high amounts of IL-10 and low levels of IL-12. Arginase enzyme converts arginine into ornithine, a precursor of polyamines and collagen, thereby contributing to the production of extracellular matrix.

When the wound-healing process is dysregulated, alternatively activated macrophages can be detrimental to the host, by leading to tissue fibrosis. Indeed, it was demonstrated that macrophages that lacked expression of IL-4 receptor failed to induce this pathology, and treatment with antibodies specific for IL-4 induced a reduction in fibrosis and a decrease in accumulation of wound-healing macrophages [21].

However, the M1-versus-M2 classification is insufficient to describe macrophages activation programs and phenotypes that can be driven by different stimuli in the microenvironment. This is of exceptional relevance in the liver, where the macrophages respond to various cues in the tissue microenvironment and can switch from a pro-inflammatory to a pro-repair phenotype. In fact, hepatic macrophages often express typical M1 and M2 markers simultaneously [13].

### 2.4. Macrophages: Central Regulators of Inflammation and Fibrosis

#### 2.4.1. Macrophages in Tissue Injury and Inflammation

Macrophages responding to infections and sterile tissue injuries are activated by inflammatory signals (e.g., PAMPs and DAMPs) in their microenvironment and polarize into classical activated M1 macrophages.

M1 macrophages synthesize and release a myriad of pro-inflammatory and cytotoxic molecules, including cytokines (e.g., IL-1, IL-12, IL-23, TNF-α), chemokines, ROS, and RNS that are responsible to amplify the inflammation and attract other immune cells, such as neutrophils and NK cells. Further, M1 macrophages have enhanced phagocytic abilities and increased the expression of co-receptors required for antigen presentation [25].

In particular, as shown in Table 2:➢TNF-α and IL-1 upregulate adhesion molecules and stimulate the endothelium to produce chemokines. TNF-α also sensitizes neutrophils and macrophages to produce ROS and RNS, and, along with IL-1, it induces the release of pro-inflammatory mediators including IL-6, platelet-activating factor (PAF), prostaglandins, matrix metalloproteinase, and various chemokines from macrophages and other cell types [26].➢Chemokines have been characterized in two major structurally distinct groups: C-C chemokines, which induce migration and activation of macrophages/monocytes and lymphocytes, and C-X-C chemokines, which are primarily neutrophil chemoattractants and activators. Continuous local release of chemokines at sites of injury is thought to mediate the ongoing migration of effector cells into inflammatory lesion [26].➢ROS and RNS (e.g., superoxide anion, hydrogen peroxide, hydroxyl radical, nitric oxide, and peroxynitrite) are produced in high amounts by macrophages via enzyme-catalyzed reactions. These reactive species are needed to destroy invading pathogens and foreign materials. Specifically, ROS are generated via membrane-associated NADPH oxidases. This enzyme produces superoxide anion that rapidly dismutates to hydrogen peroxide anion and then in presence of transition metals forms hydroxyl radicals. RNS are generated by inducible nitric oxide synthase-2 (iNOS-2) that catalyzes the oxidation of L-arginine to nitric oxide and citrulline. Nitric oxide reacts rapidly with superoxide anion to form peroxynitrite, a relatively long-lived cytotoxic oxidant.

A persistent pro-inflammatory phenotype of macrophages is responsible to turn acute into chronic inflammation and loss of tissue that leads to a variety of chronic inflammatory and autoimmune diseases.

#### 2.4.2. Macrophages in the Resolution of Inflammation

Once the acute injury has been controlled, macrophages play a role in suppressing inflammation and initiating wound repair by clearing debris and producing growth factors and mediators that provide trophic support to the tissue in which they reside [32]. For the resolution of inflammation, the “waste elimination process” is fundamental. It consists in the removal of dead cells, in particular neutrophils that undergo apoptosis and are taken up by macrophages, in the limitation or cessation of monocytes infiltration in the injured sites, and in avoiding the persistent exposure of all immunostimulatory elements to immune cells. A key regulator of this process might be the circulating serum amyloid P that is responsible to opsonize dead cells in damaged tissue.

During this phase, macrophages switch their phenotypes from classical-activated M1 macrophages to alternatively-activated M2 macrophages. M2 macrophages secrete several anti-inflammatory mediators, such as IL-4, IL-13, IL-10, and TGF-β. Other pro-fibrotic factors are the connective tissue growth factor (CTGF), CCL17, CCL22, and Igf1 [25]. Then, M2 macrophages can directly promote fibrogenesis by activating expression of arginase, the enzyme responsible for the synthesis of glutamate and proline which are in turn necessary for the synthesis of collagen [25]. Injection of M2 macrophages into mice has been demonstrate to be protective in terms of inflammatory cytokine expression and accumulation of pro-inflammatory macrophages [25]. Further, depletion of M2 macrophages from sterile wounds not only delays wound healing but also leads to apoptosis of endothelial cells [25].

Therefore, the anti-inflammatory phenotype of macrophages is mandatory for an efficient resolution of inflammation.

#### 2.4.3. Macrophages in the Resolution Process and Fibrosis

The process for the resolution of tissue damages is subdivided into regeneration and repair. Regeneration consists in the proliferation of cells to replace the injured tissues and loss structures and leads to a completely reconstitution of the lost or damaged tissue. Differently, in the repair process, the replacement of damaged tissues is incomplete, and can result in structural derangement, characterized by formation of a scar. The contribution of regeneration and scarring depends on the regenerative capacity of the tissue and the severity and nature of the injury. For example, liver and skeletal muscles have high regenerative ability and a complete functional regeneration of tissues can be obtained through regeneration of parenchymal cells [33]. On the contrary, brain and heart have limited regeneration capacity and the healing proceeds rapidly through processes of wound closure and fibrotic scarring at the expense of tissue structure and function [33]. When the injury is severe and persistent, regeneration is not possible, and an insufficient resolution results in fibrosis and dysfunction of the tissue. This is associated with the presence of alternatively-activated M2 macrophages that contribute to tissue fibrosis, through the production of several growth mediators, supporting mesenchymal healing response, fibroblasts activation, and ECM secretion.

Understanding the role of macrophages in orchestrating all phases of tissue regeneration in health and disease might be an attractive therapeutic target to limit both scarring and fibrosis.

## 3. Macrophages and Liver Fibrosis

Worldwide, chronic liver diseases (CLDs) are a major cause of morbidity and mortality, with 2 million individuals dying of liver disease each year [34,35,36]. In this context, CLDs are the 10th cause of death worldwide driven by chronic hepatitis B virus (HBV), hepatitis C virus (HCV), nonalcoholic fatty liver disease (NAFLD), and alcoholic liver disease (ALD). Mortality from CLDs in the UK has increased by over 400% since 1970 [37]. In addition to mortality, CLDs lead to a significant impairment of patients’ quality of life [35], as it is associated with life-threatening complications, including liver failure, infections, portal hypertension with esophageal and variceal bleeding or ascites, and a 2 to 6% annual incidence of primary liver cancer [38].

Liver fibrosis is a common feature of CLDs. It is the result of the wound-healing response of the liver to repeated injury [39,40]. Following a persistent liver injury, fibrillar collagen and other extracellular matrix components accumulate to high levels and determine advanced fibrosis or cirrhosis. Ominous consequences of liver cirrhosis are liver failure and development of portal hypertension due to the increased intrahepatic resistance, requiring liver transplantation to prevent liver-related death. In the United States population, intermediate to high liver fibrosis scores were associated with increased liver disease mortality [41]. In advanced stages, the liver contains approximately six times more ECM than normal, including collagens (I, III, and IV), fibronectin, undulin, elastin, laminin, hyaluronan, and proteoglycans [39]. The accumulation of ECM derives from both increased synthesis and decreased degradation [39], and most ECM components are produced by hepatic stellate cells (HSCs) [42]. HSCs have numerous functions, e.g., vitamin A storage, hemodynamic functions, immunoregulation, and extracellular matrix (ECM) remodeling. Upon liver injury, they transdifferentiate into myofibroblast-like cells, and acquire contractile, pro-inflammatory, and fibrogenic function (Figure 3).

Recent studies have revealed that hepatic resident population of KCs tightly regulate the pathogenesis of fibrosis, also at early stage of liver damage, through the release of reactive oxygen species (ROS), pro-inflammatory cytokines and eicosanoids. Titos et al., showed that in animals with experimental liver disease, the number of macrophages was increased consistently and correlates closely with the degree of hepatic injury [43] (Figure 4). KCs constitute 80–90% of the tissue macrophages in the reticuloendothelial system, account for approximately 15% of the total liver cell population [44], are resident, self-renewing and non-migrating macrophages [12] (Figure 3 and Figure 4). They are almost always found in close proximity with collagen-producing myofibroblasts and there is strong evidence that this interaction is reciprocal [42]. Activated HSCs attract and stimulate macrophages through the release of several chemokines and macrophages colony-stimulating factors (M-CSF). In turn, KCs are one of the major producer of profibrotic mediators, such as the transforming growth factor-beta1 (TGF-β1) and the platelet-derived growth factors (PDGF) that activate HSCs. Kupffer cells-derived TGF-β1 directly activates fibroblasts, promotes HSCs differentiation into myofibroblasts, enhances the expression of tissue inhibitors of matrix metalloproteinases (TIMPs) and recruits additional inflammatory cells in the injured site. TGF-β1 is considered the main cytokine that drives fibrosis in various animal models of hepatic damage, including alcoholic liver fibrogenesis, schistosomiasis and CCl4-induced fibrosis, and one of the major factors involved in fibrosis in patients with chronic liver disease [44].

Additionally, KCs remove death hepatocytes and other cellular debris by phagocytosis and, interestingly, KCs can be distinguished in “large Kupffer cells” that are predominantly localized in the periportal zone, with increased phagocytic capabilities, and in “small Kupffer cells” located mainly in the midzonal and perivenous regions, with a stronger capacity for the production of cytokines and chemokines [45,46].

KCs secrete a variety of chemokines to recruit monocytes and other leukocytes. KCs are a major source of CCL2, which recruits CCR2^+^ monocytes into the diseased liver. KCs also secrete CXCL1, CXCL2, and CXCL8 to attract neutrophils, which contribute to hepatic ischemia/reperfusion injury and heat-induced liver injury [9]. Further, KCs secrete gelatinase B, also known as matrix metalloproteinase-9 (MMP-9). MMP-9 is involved in the ECM degradation and increases matrix permeability and chemotactic ECM fragments, thereby enhancing leukocyte infiltration and inflammation leading to impaired liver function [47]. Geervliet and Bansal reported that MMP-9 inhibitors reduced leukocyte infiltration, hence inhibiting liver inflammation and damage, and the inhibition of MMP-9 expression via iNOS deficiency attenuated leukocyte infiltration, inflammation and liver damage [47].

Depletion of KCs by treatment with liposomal clodronate [48], gadolinium chloride [49], or by a unique conditional ablation system mediated by the diphtheria toxin receptor [50] has evidenced the crucial role of KCs in hepatic damage provided by attenuating hepatotoxicity in terms of steatosis, inflammation, necrosis, and collagen content in animals [43].

It has been widely described macrophage roles in liver fibrosis, acute liver failure (ALF), non-alcoholic fatty liver disease (NAFLD), alcoholic liver disease (ALD), viral hepatitis and hepatocellular carcinoma (HCC). Developments in understanding KCs biology, and therefore the identification of pathways that regulate their recruitment, activation, and polarization, can provide new perspectives toward the effective treatment of liver diseases.

Healthy liver parenchyma is constituted by epithelial cells (hepatocytes) that are the major parenchymal cell type and account for around 80% of the cells in the liver, and non-parenchymal cells: endothelial cells, HSCs, and KCs [51,52]. Briefly, hepatocytes are responsible for several liver functions, e.g., metabolism of toxic substances, glucose and bile synthesis. Endothelial cells line the sinusoids (fenestrated sinusoidal endothelium) and have a high permeability differently from ordinary vascular endothelial cells. HSCs, previously called Ito cells, lipocytes, perisinusoidal cells or fat-storing cells, account for 15% of total liver cells, and are mainly located in the so-called Space of Disse that contains a low density basement membrane-like matrix, separating hepatocytes from sinusoid, and therefore ensures metabolic exchange [53,54]. Physiologically, HSCs are in a resting state, and responsible for the retinoic acid (vitamin A) storage. KCs are the resident mononuclear phagocytes, scattered in the sinusoids, and play a crucial role in maintaining liver homeostasis through the clearance of senescent erythrocytes, cellular debris, pathogens, iron, and lipid metabolism. They are responsible to promote immune tolerance, as KCs constitute a significant population of antigen presenting cells. The normal subendothelial ECM is an elaborate cross-linked network of multiple proteins (e.g., such as collagens, elastins, fibronectins, laminins) and composition is fundamental to maintain all the different function of the liver cells and tissue homeostasis [55]. A plethora of signals can cause matrix and cellular alterations and, therefore, determine the initiation and progression of liver disorders, such as the following: the release of PAMPs (e.g., LPS, lipoteichoic acid (LTA), and β-glucan; the release of DAMPs by damaged/dead hepatocytes (e.g., high mobility group box 1 (HMGB1), mitochondrial DNA (mtDNA), and ATP); the release of reactive oxygen species (ROS) by damaged/dead hepatocytes; an hypoxic environment (e.g., due to the enhanced expression of hypoxia-inducible factor (HIF)-1α) [56]. All the signals lead to the activation of HSCs and KFs. Activated HSCs acquire a myofibroblast-like phenotype, proliferative, and migratory properties, and secrete large amount of ECM components in the Space of Disse [57]. The accumulation and deposition of ECM components in the Space of Disse determines the loss of hepatocyte microvilli and endothelial pores and consequently results in thickening of septa. This leads to the dysregulation of the metabolic exchange and portal hypertension.

Liver fibrosis is the common outcome of chronic liver diseases induced by viral and helminthic infections, autoimmune, metabolic, and genetic conditions, exposure to toxic compounds (e.g., acetaminophen and ethanol), and ionizing radiations (e.g., radiotherapy in cancer patients). Hepatocytes, HSCs, KCs, and other immune cells recruited in the damaged area from the blood circulation (e.g., neutrophils, monocyte derived macrophages) have been identified in the pathogenesis of liver fibrosis. Damages to hepatocytes determine oxidative stress, pro-inflammatory cascades, additional injuries to the parechymal, and amplify KCs and HSCs pools. Damaged hepatocytes and activated KCs sustain a massive accumulation of circulating immune cells in damaged areas. CCL2, CCL5 and CXCL1, released by KCs and HSCs, are considered the major drivers of MoMϕs recruitment and infiltration into the liver. In turn, infiltration of MoMϕs contribute to the expansion of hepatic macrophages.

## 4. Innovative Approaches to Target Macrophages

To date, there is no specific treatment tackling liver fibrosis *per se*: the ideal antifibrotic therapy would be one that is effective in slowing down the fibrosis process, well tolerated when administered chronically and liver-specific. The removal of the etiopathological agent still remains the most effective intervention in the treatment of liver diseases, however, this approach is not always achievable, e.g., in patients with cirrhosis and clinical complications [39]. Further, traditional treatment methods have some problems, such as toxic and side effects on tissues and organs, as well as low drug selectivity that cannot provide an effective concentration of therapeutic drugs into the liver [57].

Safe and effective antivirals can cure chronic hepatitis C. The therapeutics for chronic hepatitis B consist in the constant viral suppression with nucleoside and nucleotide drugs. In the case of NAFLD, no specific drug has been approved and the current standard management strategy moves around diet, exercise and treatment of the metabolic syndrome components. Specifically to non-alcoholic steatohepatitis (NASH), a form of NAFLD characterized by inflammation, oxidative stress, hepatocellular damage, and steatosis, most of the clinical treatments focus on inflammation, metabolism, and fibrosis [58]. Anti-inflammatory drugs for NASH include ASK-1 inhibitors, CCR-2 and CCR-5 antagonists (cencriviroc); metabolism-related drugs include lipid metabolism-related drugs, like FXR agonists (obeticolic acid), THR-β receptor agonists, and glucose metabolism-related drugs, like SGLT-2 inhibitors. However up to date there is no drug approved by the FDA for NASH treatment [58]. In the case of autoimmune hepatitis (AIH), the use of corticosteroids and immunosuppressive drugs is the standard therapy since inflammation precedes and sustains the progression of fibrosis [59]. Steroids easily bypass biological barriers and therefore are widely distributed and bioavailable after an oral administration. Patients on chronic treatment with steroids often experience several steroid-related side effects (e.g., obesity, mood and cognitive disorders, osteopenia, hypertension, and diabetes). It is urgent to find new solutions to reduce these treatment-related complications.

Several strategies have been attempted to find novel strategies to target hepatic macrophages for the treatment of liver diseases [60]:-dampening KCs activation by modifying the gut-liver axis, e.g., administration of probiotics, antibiotics or fecal microbiome;-inhibiting Ly-6C^+^ inflammatory monocyte recruitment to damaged liver, e.g., through pharmaceutical antagonists of the chemokine CCL2;-modulating hepatic macrophage functions, by delivery drugs loaded in specific carriers that can be influence the polarization of macrophages or reduce their inflammatory responses;-augmenting the differentiation to restorative macrophages.

Even more to date, well-defined drug nanocarrier systems (e.g., nanoparticles, nanoliposomes, nanomicelles, and nanohydrogels) with an excellent biocompatibility and safety are emerging as innovative and promising alternative to standard therapies for the fibrotic diseases. These systems allow achieving high concentrations of the drug into the fibrotic organ and effectively into the targeted cells.

Several nanomedicine-based therapies have been explored to cure hepatic fibrosis by delivering drugs to macrophages. Recently, for the potential treatment of NASH, J. E. Zhou et al., improved liver function and steatosis in vivo mice through the administration of a mannose-modified HMGB1-siRNA loaded stable nucleic acid lipid particle delivery system for targeting liver macrophages with mannose receptor mediation and thereby silencing HMGB1 [58]. J. Cui et al., suppressed Nogo-B expression (namely Reticulon 4B, an endoplasmatic reticulum protein releases by non-parechymal cells, e.g., KCs) by using siRNA-loaded nanoparticles. Nogo-B promotes the progression of fibrosis [61]. L. Kaps et al., demonstrated that nanohydrogel particles equipped with mannose residues on the surface delivered siRNA efficiently to M2 polarized macrophages, as they overexpress the mannose receptor CD206 [62]. L. Kaps et al., developed pH-degradable, squaric ester-based nanogel carries system for the delivery of the bisphosphonate, an agent that reprograms profibrotic M2- toward antifibrotic M1-macrophages. They demonstrated that this approach potently prevented liver progression in vivo [38].

Here, we decided to analyze in more details other innovative therapeutic strategies recently developed to target hepatic phagocytic cells.

### 4.1. ANANAS Liver: Nanoassemblies

ANANAS formulation is based on the Avidin-Nucleic-Acid-Nano-ASsemblies platform and consists in poly-avidin nanoparticles that form upon the high affinity driven nucleation of avidin units around a non-coding plasmid DNA [63]. ANANAS is made of soft biodegradable and biocompatible components and is poorly immunogenic, since the avidin in the assembly does not induce abnormal immune response in healthy NFR mice [63]. In the same mouse model, ANANAS formulation possesses some features that make it suitable as a drug carrier to treat a broad range of liver disorders, as many nanoparticle (NP)-based carriers accumulate at the liver due to the filter functions of this organ and independently of the presence of liver targeting moieties [63]. Furthermore, ANANAS nanoparticles possess a high stability in the bloodstream and a pH dependent release into the target.

Morpurgo et al., formulated a dexamethasone-carrying nanoformulation based on ANANAS nanoparticles [64]. Reversible biotinylated drugs, e.g., dexamethasone (Dex), can bind to the avidin biotin binding sites (BBS) and therefore be located at the nanoparticle surface (ANANAS-Hz-Dex). They showed that this drug delivery system displayed strong tropism for the liver without the need for a liver targeting element [64].

In the well-established cytochrome P450 2D6 (CYP2D6) mouse model that reflects many aspects of human AIH [65], they demonstrated that ANANAS-Hz-Dex nanoassemblies selectively entered in liver KCs lysosome upon intraperitoneal administration. KCs have a huge intrinsic ability to recognize, and digest particles and other foreign material and this peculiar ability make them a perfect target by nanomaterials of different nature and the lower pH in lysosomes allows the breakage of the linker and the selective release of Dex inside the target cells. Further, they demonstrated that this nanoformulation did not release the dexamethasone in any body district other than the liver and ANANAS-Hz-Dex nanoformulation were more effective than the free drug in controlling the disease in the animal mode.

These results make this system an extremely interesting tool for future clinical trials in patients with liver diseases.

### 4.2. Microparticles Delivery to Macrophage Cells

Phagocytosis process might be exploited to target macrophage cells. Phagocytosis is defined as the receptor-mediated engulfment of large particles (>0.5 μm) into plasma membrane-derived vacuoles called phagosomes [66]. Only specialized cells, termed “professional phagocytes”, are capable to perform this process with high efficiency. Neutrophils, macrophages, dendritic cells, and monocytes are some of these cells. Phagocytes recognize a large number of different particles that can be ingested, including all sorts of microbial pathogens, apoptotic cells, and foreign substances. Macrophages can ingest large particles, whereas all other non-phagocytic cells are not able to uptake >0.5 μm particles, and this is considered a potential way to target macrophages and thus develop a macrophage-specific drug-targeting strategy.

Binatti et al. [67] investigated whether astaxanthin, an antioxidant and anti-inflammatory compound [68], encapsulated into micrometer-sized particles could help control macrophage activation in radiation-induced fibrosis (RIF). RIF is a serious long-term side effect that compromises respiratory, hepatic, intestinal, and urinary functions in cancer patients exposed to radiotherapy [69]. Its pathogenesis is still not fully understood, although it is known that macrophages have a central role in RIF for their ability to produce reactive oxygen species (ROS), and inflammatory molecules that in turn trigger the fibrotic pathways in injured tissues [70]. They showed [67] in vitro experiments that these microparticles were not cytotoxic for macrophage cells and could be taken up only by them. Their results indicated that phagocytosis of astaxanthin-loaded microparticles protected macrophages from H_2_O_2_-induced oxidative stress through the reduction of intracellular ROS accumulation by ~75% of control untreated cells. Astaxanthin-loaded (ASX) microparticles inhibited the production of bioavailable TGF-β by interfering with the activity of intracellular convertases, a class of enzymes implicated in the post-translational processing and activation of numerous pro-proteins (e.g., TGF-β). Further, they demonstrated that the effects of astaxanthin-loaded microparticles could be further potentiated by combination treatment with pentoxifylline [70], a drug approved for the treatment of claudication as it has anti-inflammatory properties (e.g., inhibition of TNF-α production and signaling) and improves rheological properties of blood and wound healing [71]. Given in combination with ASX microparticles, that on their own inhibit active TGF-β release by targeted macrophages, might result in more effective treatments against inflammation and fibrosis.

Although within the limitations of a pre-clinical in vitro study, this treatment strategy developed to target specifically phagocytic cells might be effect for the treatment of fibrotic conditions, in particular for liver fibrosis, as ASX has been demonstrated in rat liver to have hepato-protective effects in fibrotic conditions [72,73,74]. It reduces oxidative stress, suppresses the upregulation of fibrotic genes, regulates macrophages homeostasis, and prevents and reversed the activation of mouse primary HSCs.

### 4.3. GKT137831, an Oral NOX1/NOX4 Inhibitor

Chronic liver diseases, including liver fibrosis, cholestatic liver injury, NASH, alcohol-induced hepatitis, viral hepatitis, and HCC, account for nicotinamide adenine dinucleotide phosphate oxidase (NOX or NADPH) activation (*via* phosphorylation) that occurs as a consequence of chronic damages to the liver. NOX activation represents a unique intracellular amplifier of several signaling pathways that account for excessive tissue remodeling and the subsequent fibrosis [75]. NOX is an enzyme system, a multicomponent complex of proteins [75,76] that catalyze reactions to generate radical oxygen species (ROS), by transporting electrons through the cell membrane. In humans, seven isoforms for NOX family have been identified, including NOX1-5, dual oxidase 1, and dual oxidase 2. NOX 1, NOX 2, and NOX 4 are the major members expressed in the liver, specifically in KCs and HSCs. Specifically NOX1 and NOX2 are critical for the differentiation of monocytes to macrophages, the polarization of M2-type. Xu et al., showed that the loss of NOX1 and NOX2 effectively reduced the NADPH activity and ROS produced by macrophages infiltrating the wound site and further inhibited M-CSF-stimulated M2 macrophage differentiation [77]. Indeed, NOX1 and NOX2 were critical for the activation of the MAPKs JNK and ERK during macrophage differentiation and that the deficiency of JNK and ERK activation was responsible for the failure of monocyte-to-macrophage differentiation, in turn affecting M2 macrophage polarization [78]. They found that deletion of both NOX1 and NOX2 led to a dramatic decrease in ROS production in macrophages and resulted in impaired efficiency in monocyte-to-macrophage differentiation and M2-type macrophage polarization [77].

As NOXs are implicated in many cellular processes that lead to an aberrant tissue repair, pharmaceutical targets for NOXs could provide an anti-fibrotic approach. GKT137831, a small organic molecule of low molecular weight and a member of the pyrazolopyridine dione chemical class, is a selective inhibitor of NOX 1 and 4 and represents the first drug in the class of NOX inhibitors to enter the clinic. It is being investigated in several fibrotic disorders including Primary Biliary Cholangitis (PBC), NASH, Idiopathic Pulmonary Fibrosis (IPF), Diabetic Kidney Disease (DKD), Systemic Sclerosis (SSc), as well as in Head and Neck cancer. Aoyama et al., reported that GKT137831 showed significant improvement of liver fibrosis by suppressing ROS production, via NADPH inhibition, and fibrotic gene expression in mice induced by carbon tetrachloride (CCL4) and duct ligation [79,80]. In a phase 2 clinical trial for patients with PBC (NCT03226067), GKT137831 showed significant effects on serological cholestasis parameters [78,79,81]. NOX1/4 inhibition with GKT137831 may represent an attractive therapeutic strategy for fibrotic disorders.

(For a complete overview of the advances in the research of nanocarriers for targeted treatments of liver fibrosis, see references [57,82]).

## 5. Conclusion and Future Perspectives

Macrophages are widely distributed throughout the tissues where they hold a central position to maintain homeostasis and have protective roles. However, in pathological conditions, they acquire harmful functions behaving as pathogenic drivers for inflammation and fibrosis. Expanding knowledge on the large spectrum of macrophage phenotypes, their heterogeneity, plasticity, and their different functions has given the opportunity to develop new compounds for the treatment of fibrotic disorders. Several approaches have been attempted to specifically target macrophage cells to reduce their responses to injuries. It is crucial to target the pathogenic subset without hindering the homeostatic component, thus avoiding an unwanted immune suppression.

In this review, we reported the most recent strategies developed in pre-clinical in-vitro and in-vivo models, in particular targeted therapies based on the reduction of phagocytic cells activation and their inflammatory and pro-fibrotic responses. This type of strategy represents an area of great interest and rapid advancement, as macrophages are phagocytic cells that can internalize nano- and microparticles at higher efficiency than other cells, potentially minimizing side effects. Another innovative strategy is to inhibit NADPH oxidase enzymes that are mostly expressed in macrophages and increase oxidative stress by generating superoxide and hydrogen peroxide from molecular oxygen.

However, there are still several challenges retarding drug development and its translation to clinic practice: limitations of murine models; the greater heterogeneity of macrophages in humans compared to animal models; the disparity in macrophage phenotypes between humans and mice; less knowledge of cellular subsets in humans compared to mice; technical difficulties to isolate liver macrophages; the variability in timing and dosing to achieve therapeutic effects. All these challenges need to be addressed before moving to clinical trials in humans.

## Figures and Tables

**Figure 1 ijms-23-06649-f001:**
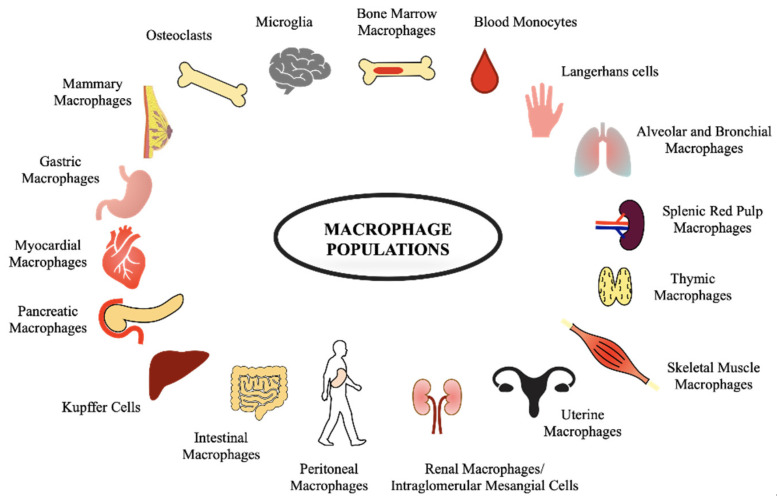
Examples of different macrophage populations associated with different tissues are illustrated.

**Figure 2 ijms-23-06649-f002:**
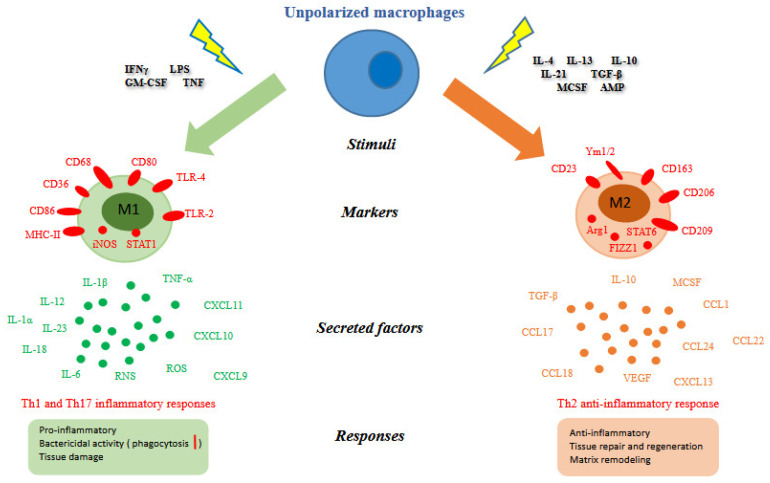
Macrophage polarization. Different stimuli, surface and intracellular markers, secreted factors and biological responses [22,23,24] between M1 and M2 macrophages were summarized. Abbreviation: IFNγ (interferon gamma); LPS (lipopolysaccharide); GM-CSF (granulocyte-macrophage colony stimulating factor); TNF (tumor necrosis factor); CD (cluster of differentiation); MHC-II (major histocompatibility complex class II); iNOS (inducible nitric oxide synthase); STAT (signal transducer and activator of transcription); TLR (toll-like receptor); Arg1 (arginase 1); FIZZ1 (resistin-like molecule alpha1); IL (interleukin); TGF-β (tumor growth factor-beta); MCSF (macrophage colony-stimulating factor); AMP (cyclic adenosine monophosphate); CCL (chemokine C-C motif ligand); CXCL (C-X-C motif ligand); ROS (reactive oxygen species); RNS (reactive nitrogen species); VEGF (vascular endothelial growth factor).

**Figure 3 ijms-23-06649-f003:**
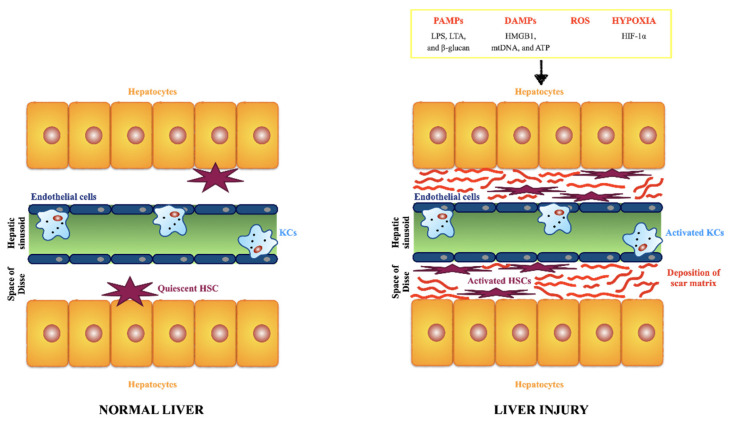
Comparison of matrix and cellular environments in healthy and damaged liver.

**Figure 4 ijms-23-06649-f004:**
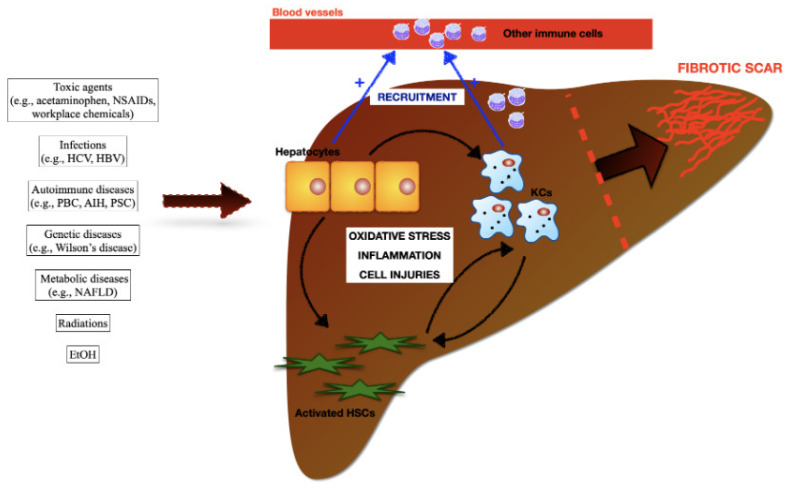
Schematic representation for the development of fibrotic scar in the injured liver.

**Table 1 ijms-23-06649-t001:** Different phenotypes and functions of resident macrophages in the main organs of the body. Abbreviations: Clec4f (C-type lectin domain family 4, member F); CD68 (cluster of differentiation 68); CD14 (cluster of differentiation 14); CR3 (complement receptor 3); CD206 (cluster of differentiation 206); SR-A (scavenger receptor-A); MARCO (macrophage receptor with collagenous structure); CD11 (cluster of differentiation 11).

Surface Markers	Pathology	Physiological Functions	Organ
Murine: F4/80^+^; Clec4f^+^; Human: CD68^+^; CD14^+^	-Impaired erythrocyte turnover and Fe recycling -Uncontrolled inflammation -Fibrosis	-Hepatic tissue remodeling -Erythrocyte turnover, Fe recycling, bilirubin metabolism, and clearance of thromboplastins, fibrin, aggregated platelets -Lipid metabolism: uptake and catabolism of lipoproteins, metabolism of cholesterol and sterols -Clearance of bacterial, parasites, viruses, toxins, cellular debris from blood	Liver
F4/80 ^dim^; CR3^dim^; CD206^+^; MARCO^+^; SR-A^+^	-Alveolar proteinosis -Inflammation -Fibrosis	-Surfactant metabolism -Inhaled particles clearance (phagocytosis) -Immune surveillance against pathogens, and pollutants	Lung
CD11b^+^; F4/80^+^; CD206	Impaired erythrocyte turnover and Fe recycling	-Clearance of dysfunctional or old erythrocytes -Haem catabolism -Fe recycling	Spleen
F4/80^+^, CR3^+^	Neurodegeneration	-Elimination of old and dead neurons -Synaptic remodeling -Immune surveillance -Brain development	Brain
CD11b^+^; F4/80+	-Heart block -Arteriosclerosis	Regulate cardiomyocyte electrical activity through macrophage Connexin43- mediated adhesion -Surveillance	Heart and vasculature

**Table 2 ijms-23-06649-t002:** Schematic description of macrophage contribution in oxidative stress, inflammation, and fibrosis. Abbreviations: ROS (reactive oxygen species); RNS (reactive nitrogen species); NADPH (nicotinamide adenine dinucleotide phosphate); TNF-α (tumor necrosis factor-alpha); IL-1β (interleukin-1beta); IL-12 (interleukin-12); IL-6 (interleukin-6); CCL-2 (chemokine C-C motif ligand-2); TGF-β (tumor growth factor-beta); EMT (epithelial mesenchymal transition); ECM (extracellular matrix).

Items	Description
**Oxidative stress**	Activated macrophages are the major responsible for the oxidative stress. ROS and RNS recombine with macromolecules and lead to the degradation of protein, lipid peroxidation, and oxidation of DNA. NADPH, also called NOX, is the key enzyme for ROS production (e.g., superoxide, hydroxyl and peroxyl radicals). Inducible NO synthase (iNOS) is a key enzyme for RNS production (e.g., nitric oxide and peroxynitrite).
**Inflammation**	Activated macrophages secrete pro-inflammatory factors. - cytokines: TNF-α; IL-1β; IL-12; IL-6 TNF-α is produced predominantly at sites of inflammation by activated monocytes and macrophages. TNF-α induces hepatocytes apoptosis, and activation of HSCs. In chronic inflammatory hepatopathies, IL-1β is mostly produced by KCs, hepatocytes, and adipocytes. In KCs, it promotes liver inflammation by amplifying inflammasome activation, TNFα, and IL-β release. IL-1β sustains hepatic steatosis by stimulating triglyceride and cholesterol accumulation in primary liver hepatocytes, lipid droplet formation, and insulin resistance in hepatocytes [27,28]. IL-6 dependent signaling in the liver is critical for the induction of the acute-phase response [29]. The main role of IL-12 is to stimulate immune cells and other lymphocytes. - chemokines regulate the migration and functions of KCs, HSCs, hepatocytes, endothelial cells, and the recruitment of circulating immune cells. *CCL2, also known as monocyte chemotactic factor 1 (MCP1),* is one of the best characterized chemokines in the hepatic fibrogenesis. CCL2 regulates the migration and infiltration of monocytes/macrophages through combination with its specific receptor CCR2 [30]. It has been reported that CCL2/CCR2 axis has a vital role during fibrosis [30].
**Fibrosis**	Activated macrophages produce high amount of TGF-β [31]. TGF-β sustains oxidative stress, proliferation and activation of HSCs into myofibroblast-like cells, proliferation and differentiation of fibroblasts into myofibroblast, EMT, synthesis of ECM and stimulate the expression of metalloproteinase inhibitors.

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
