# Peer review of "The Role of Macrophages in Liver Fibrosis: New Therapeutic Opportunities"

_ijms, 2022, doi:10.3390/ijms23126649_

Round 1
Reviewer 1 Report
The review of Binatti et al is globally well structured and addresses issues that might be of general interest like the pharmacological targeting of macrophages. However, I have some points that need to be resolved before acceptance of the review for publication:
1. English proofreading must be carried out. There many sentences that need to be rephrased/restructured and words that sometimes are misspelled. Here are some examples (non exhaustive):
- Line 45 “process” must be changed to “processes”
- Line 46 the verb “ultimate” must be replaced
- Lines 56-57 this sentence has many mistakes: “know” is probably “now”, “approved of efficacious”… approved efficient…
- Lines 58-63 Too long, please, simplify.
- Line 201 “antigenic” to be changed to “antigen”
- line 223 “alternative” to be changed to alternatively
- line 226 “determine” to be replaced
- line 310 “replace” to be changed to “replacement”
- line 313 “muscle” should be in plural
- lines 311-316 phrase is not clear
- lines 415-419 rephrase
- line 503 “that can” BE “ingested”
- etc.
2. Lines 67-68: what do you mean by “almost every tissue compartment”? Be more precise. Do you mean organ?
3. Line 77. What are the stages of tissue damage, repair, and fibrosis or you mean these processes separately? This is not clear.
4. Table 1 should be reformatted (this is valid for all tables) since texts from columns are merged. In table 1 by “alveolar proteinasis” did you actually mean pulmonary alveolar proteinosis?
5. Table 2. What do you mean by “IL-1beta release in KC”. It is released IN or FROM the cells?
6. I would recommend a third table to be included in which a comparison between M1 and M2 macrophages to be done (secreted cytokines, activation, function…).
7. Figures S1 and S2 should be placed within the main text and not in supplementary.
8. Line 169 “chronic insults”. Did you mean damage?
9. Lines 194-195: You say that IFN gamma is the main product of Th1 cells and just below, lane 196 you say that NK cells are the principal source of IFN gamma… This is confusing.
10. Lines 328-337. This paragraph is confusing. Is liver fibrosis considered as a CLD? Or What CLD includes? You say that 1.5 M die from cirrhosis and 2 M from liver disease each year. Does this include HCC? Ref.33 is too old. Ref. 34 to be changed. You can check PMID: 34136143, and others. I have doubts that ref.36 is well placed here (line 337). Re-organize all the paragraph.
11. Line 348. Add ref. after “proteoglycans”.
12. Line 409. Explain why “it is not always achievable”.
ANANAS liver: mention the experimental models (cells, animals?). NP in NP-based stands for what?
Author Response
Dear reviewer,
Thank you for your revision and for your helpful suggestions. We provided our responses to your comments and believe our manuscript is much improved as a result.
- we changed /simplified the text and we corrected the errors detected by the reviewer
- We modified the reference list so as suggested by the reviewer
- We changed the format and properties of the tables
- As suggested by the reviewer, we added a new figure to illustrate macrophages polarization
- Lines 67-68: what do you mean by “almost every tissue compartment”? Be more precise. Do you mean organ?
Thank you for the suggestion to be more precise. We meant in all our body as macrophages are virtually present in all our tissues and body compartments. We modified in " almost every tissues and body compartments"
- Line 77. What are the stages of tissue damage, repair, and fibrosis or you mean these processes separately? This is not clear.
Yes it was not clear, and therefore we modified the text " macrophages orchestrate the wound healing, the repair/regeneration process and the development of fibrosis". We meant separately, as macrophages initiate the inflammatory response after a tissue injury and sustain it. Then depending on the nature of the injury, macrophages change their phenotype and can resolve the injury or determine the development of fibrosis. Macrophages have a central roles in all these process, inflammation, repair/regeneration and fibrosis through the release of different meadiators.
- Table 2. What do you mean by “IL-1beta release in KC”. It is released IN or FROM the cells?
We meant that in KCs, TNF stimulates the release of IL 1beta from these cells. We modified the text
- Figures S1 and S2 should be placed within the main text and not in supplementary.
Thank you for the suggestion. We placed all the figures in the main text
- Lines 194-195: You say that IFN gamma is the main product of Th1 cells and just below, lane 196 you say that NK cells are the principal source of IFN gamma… This is confusing.
Yes it was quite confusing and we modied the text. We meant that IFN is one of the most important stimulus of M1 polarization and NKs are the main producers of IFN gamma. We rewrote the phrase
- Lines 328-337. This paragraph is confusing. Is liver fibrosis considered as a CLD? Or What CLD includes? You say that 1.5 M die from cirrhosis and 2 M from liver disease each year. Does this include HCC? Ref.33 is too old. Ref. 34 to be changed. You can check PMID: 34136143, and others. I have doubts that ref.36 is well placed here (line 337). Re-organize all the paragraph.
Thank you for the advice and we re-organized the paragraph and changed the ref. Liver fibrosis is a typical feature of CLD
- Line 409. Explain why “it is not always achievable”.
We modified the main text. Briefly, it is not achievable for patients with an advanced stage of fibrosis (cirrhosis), as the damages are no more reparable and reversible and for patients with other important complications
- ANANAS liver: mention the experimental models (cells, animals?).
Thank you for suggesting it. Therefore, we added the experimental models.
Reviewer 2 Report
Comments on Review IJMS 1743726
The Review Article IJMS 1743726 titled "The role of macorphages in liver fibrosis: new therapeutic opportunities” by Eleonora Binatti, Alessio Gerussi, Donatella Barisani and Pietro Invernizzi reports new strategies to reduce macrophages activation during the fibrotic process. First, the review starts with an overview of macrophages production differentiating the two types of population we can find upon activation and presenting their role in fibrosis. The second part of the article is centered in the specific role of these immune cells during liver fibrosis to end up with a revision of some new approaches trying to alleviate macrophages response after liver damage.
Overall the manuscript is clear and well organized, reflecting good knowledge of the topics presented. This review will be of interest for the scientific audience and should be accepted for publication. Every aspect describe is well addresses so it turns difficult to make suggestions, only minor comment to improve some parts of the text to make it easier to read and understand:
- Tables are not well tabulated and they result a bit difficult to read as all the text appear together.
- Table 2 is not clearly indicated. Some kind of heading should appear.
- The last paragraph in the Introduction section: “Here, we summarize….” Should be rewrote and shortened to facilitate understanding.
- Heading should appear just above text (avoid appearing in a different page)
Other general comments:
- Abbreviations should be revised to eliminate those only cited once. Also, all abbreviations must be explained the first time they appear:
NP-carriers (lane 473)
ASX microparticles (lane 527)
PBE (lane 573)
- Revise all “Letteting” as it seems there is more than one typology
- Some spelling mistakes in lanes: 43, 56-57, 432
Author Response
Dear reviewer,
Thank you for your revision and for your helpful suggestions. We provided our responses to your comments and believe our manuscript is much improved as a result.
-Tables are not well tabulated and they result a bit difficult to read as all the text appear together. -Table 2 is not clearly indicated. Some kind of heading should appear.
We changed the format and properties of the tables (the odt format might have changed the properties of the table if it was opened as a word document, but in any case we modified the tables to make them easier to read).
-The last paragraph in the Introduction section: “Here, we summarize....” Should be rewrote and shortened to facilitate understanding.
Thank you for the advice, and we reformulated the paragraph to facilitate understanding.
-Heading should appear just above text (avoid appearing in a different page). Thank you, we modified this mistake
Other general comments:
-Abbreviations should be revised to eliminate those only cited once. Also, all abbreviations must be explained the first time they appear.
Thank you, we modified the abbreviations. I preferred only for figures to report the abbreviation even if they are already present in the main text. And we explained new abbreviations the first time they appear.
I revised the latteting and spelling mistakes.